# Patch Vestiges in the Adversarial Examples Against Vision Transformer Can Be Leveraged for Adversarial Detection

**Juzheng Li**

Inno (Shanghai) Asset Management Co., Ltd.
Beijing, China, 100032
lijuzheng09@gmail.com

## Abstract

Vision Transformer (ViT), a Transformer-based architecture that divides images into patches, can catch up with or surpass convolution-based networks in multiple Computer Vision tasks. However, ViT is also vulnerable in the face of adversarial examples (AEs). Thus the topic around the attack and defense of ViT becomes very rewarding. Recent studies have found that the AEs against ViT seem to have grid-like textures that coincide with the patches. In this paper we confirm such sensation is true. We show that these grid-like textures are the remained vestiges due to the patch division from ViT. We name them as Patch Vestiges. We propose statistics to measure the sizes of Patch Vestiges in the images or AEs quantitatively. We also build a linear regression classifier to detect the AEs against ViT practically via the proposed statistics. The experiments show that the performance of the simple classifier can even match some recent adversarial detection methods, suggesting that when trying to attack ViT or detect the AEs against ViT, Patch Vestiges are worth considering about as a critical factor.

Transformer (Vaswani et al. 2017) is almost based on self-attention mechanisms and fully connected layers. It creatively subverts the architecture of RNNs and realizes the state-of-the-art performances on almost all Natural Language Processing tasks. It is naturally hoped that Transformer can be applied to the field of Computer Vision. However, Transformer requires a sequential input that has a quite different shape from an image. Vision Transformer (ViT) (Dosovitskiy et al. 2020) overcomes the difficulty by dividing an image into small patches and linking them into a sequence. With the help of Transformer, ViT achieves excellent performances in many Computer Vision tasks.

Although ViT is effective, it has similar weakness with CNNs in front of the adversarial examples. Adversarial Examples (AEs) (Szegedy et al. 2013) are images with artificial perturbations that are small enough to fool the human eyes but can make deep neural networks output wrong results. Some preliminary studies (Bhojanapalli et al. 2021; Shao et al. 2021; Mahmood, Mahmood, and van Dijk 2021) show that ViT is vulnerable to all common AEs, and even weaker than CNNs under some attacks. The good news is that it is difficult for the AEs against CNNs to transfer to ViT

directly (Shao et al. 2021; Naseer et al. 2021; Aldahdooh, Hamidouche, and Déforges 2021). Thus it is meaningful to study the unique natures of the AEs against ViT.

To the human eye, the magnified adversarial perturbations of the AEs against ViT seem to have grid-like textures and exhibit some periodicity and repetition (Bhojanapalli et al. 2021), as shown in Figure 1. This is the initial inspiration of this paper. A very intuitive conjecture is that the AEs against ViT may also be divided into patches. In this paper, we confirm this conjecture is true and bring up the concept Patch Vestiges. We define Patch Vestiges as the abnormalities of the AEs against ViT that are caused by the patch division.

We also find a method to measure Patch Vestiges quantitatively. We propose *Leaps* to measure the step changes between two adjacent pixels in different patches. We assume the step changes are the key points of Patch Vestiges. Additionally, we propose statistics PV, IPC and NCC based on *Leaps* and build a binary linear regression classifier on them. The experiments show that our approximations on *Leaps* are successful and that by our proposed statistics PV, IPC and NCC, the linear regression classifier can detect the AEs against ViT effectively.

We sum up the key contributions of this paper as follows:

- We substantiate the human instinct that the patches used in Vision Transformer remain vestiges in the adversarial examples.

- We bring up the concept Patch Vestiges and find a quantitative measurement for them.

- We prove that Patch Vestiges can be a critical weakness of the adversarial examples against Vision Transformer.

## Related Work

**Vision Transformers** Vision Transformer (ViT) (Dosovitskiy et al. 2020) is the first work to successfully leverage Transformer (Vaswani et al. 2017) in Computer Vision tasks by dividing images into patches. DeiT (Touvron et al. 2021) uses a similar model structure but adds a new distillation token. T2T-ViT (Yuan et al. 2021) recursively integrates the adjacent tokens to better extract the low-level image features. Recently, Swin Transformer (Liu et al. 2021) shows the superiority of Transformer and defeats CNN-based models in many tasks by bringing in the shifted window scheme.

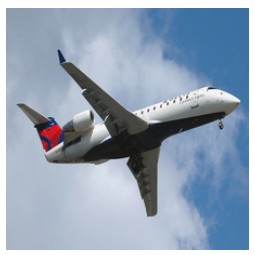 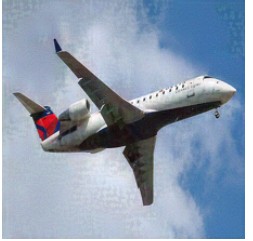 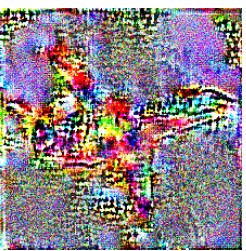 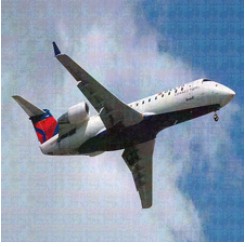 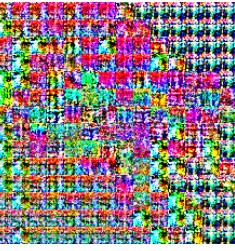

(a) ResNet / ViT "plane"  (b) ResNet AE "dog"  (c) ResNet AE perturb.  (d) ViT AE "dog"  (e) ViT AE perturb.

Figure 1: A clean image and its AEs and according adversarial perturbations. The image "plane" is chosen from the ILSVRC2012 (ImageNet) dataset (Russakovsky et al. 2015). ResNet and ViT both give the correct classification when the input is clean. The AEs generated by PGD $\ell_\infty = 8$ make ResNet and ViT output the wrong category "dog" respectively. The adversarial perturbations are effect images that are magnified from the real values to make them explicit.

**Adversarial Detection** Bayesian uncertainty (BU) and kernel density (KD) are previously proposed to detect the out-of-manifold points (Feinman et al. 2017). RCE (Pang et al. 2018) uses a new Reverse Cross-Entropy based on KD to better distance the clean images from the AEs. LID (Ma et al. 2018) detects the AEs by the local sparseness. A Mahalanobis distance based score is afterwards proposed (Lee et al. 2018). Under the assumption that AEs are out of the manifold of the natural scenes, natural scene statistics (NSS) are used in the detector (Kherchouche et al. 2020). More recently, LiBRe (Deng et al. 2021) leverages Bayesian neural networks with refined training procedures for adversarial detection.

## Methodology

Despite the recent excellent improvements of ViT, we focus on the vanilla ViT model (Dosovitskiy et al. 2020) because the fixed division makes the research stable. The vanilla ViT divides an image into $n \times n$ patches in a grid shape, making several horizontal and vertical dividing lines. Intuitively, the adversarial perturbations of the adjacent pixels astride the dividing lines should have step changes because they come from different partial differential expressions. We measure the step changes by *Leaps*. To calculate *Leaps*, we approximately assume that the pixel values of the clean images and the adversarial perturbations inside the patches vary mildly, and only the adversarial perturbations across the patches are violent, as shown in Figure 2(a).

We calculate *Leaps* as follows. We denote the change of the pixel values between the adjacent pixels $i$ and $j$ by $G(i, j)$. For both clean images and inside-patch adversarial perturbations, a center $G$ should be equal to the average of its bilateral $G$s under our approximation. But for adversarial perturbations astride the dividing lines, the equality does not hold. Thus we define

$$Leap(i, \vdash)$$
$$= \left| G(i, i \oplus 1) - \frac{(G(i, i \ominus 1) + G(i \oplus 1, i \oplus 2))}{2} \right|, \quad (1)$$

where $\vdash$ means the alternative direction that is either horizontal or vertical and $i \oplus n, i \ominus n$ means moving $n$ pixels forward or backward along the direction $\vdash$ from the pixel $i$.

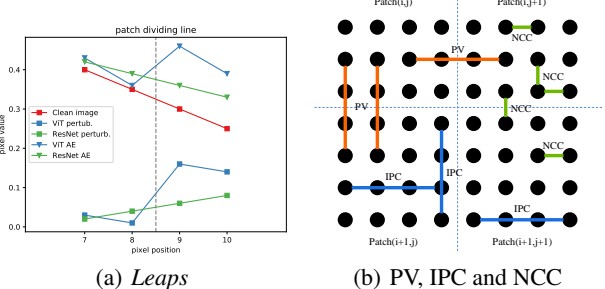

(a) *Leaps*  (b) PV, IPC and NCC

Figure 2: (a) The illustrative diagrams of *Leaps*. (b) The example positions of the proposed statistics PV, IPC and NCC.

$Leap(i, \vdash)$ shows the non-smoothness of the local changes $G$ around the pixel $i$. If $Leap(i, \vdash)$ is high and pixels $i$ and $i \oplus 1$ stride over a dividing line, there will be larger possibility that the given image is an AE against ViT.

Based on *Leaps*, we propose PV, IPC and NCC, standing for Patch Vestiges, Inside-Patch Contrast and Natural Change Contrast respectively. PV consists of *Leaps* astride the dividing lines, IPC consists of *Leaps* that are fully inside the patches, and NCC consists of all the adjacent changes (see Figure 2(b)). More precise definitions are:

$$PV(X) = Ave_{\vdash \in \{-, |\}, i \in PB(X, \vdash)}(Leap(i, \vdash)),$$
$$IPC(X) = Ave_{\vdash \in \{-, |\}, i \in PI(X)}(Leap(i, \vdash)), \quad (2)$$
$$NCC(X) = Ave_{\vdash \in \{-, |\}, i \in X}(|G(i, i \oplus 1)|),$$

where $Ave$ means the average, $PB(X, \vdash)$ means the pixel set that $i$ and $i \oplus 1$ stride over a dividing line, and $PI(X)$ means the pixel set that $i, i \oplus 1, i \oplus 2, i \ominus 1$ are in the same patch. Under this definition, PV will be much higher than IPC only for the AEs with large Patch Vestiges. NCC measures the natural pixel fluctuations of clean images and is a baseline for PV and IPC.

We also leverage linear regression and build a simple binary classifier $y = a_1 PV + a_2 IPC + a_3 NCC + a_4$, where $a_1, a_2, a_3, a_4$ are trainable parameters. If PV of the AEs against ViT is very different from IPC, the binary classifier will have high capacity to distinguish those AEs from

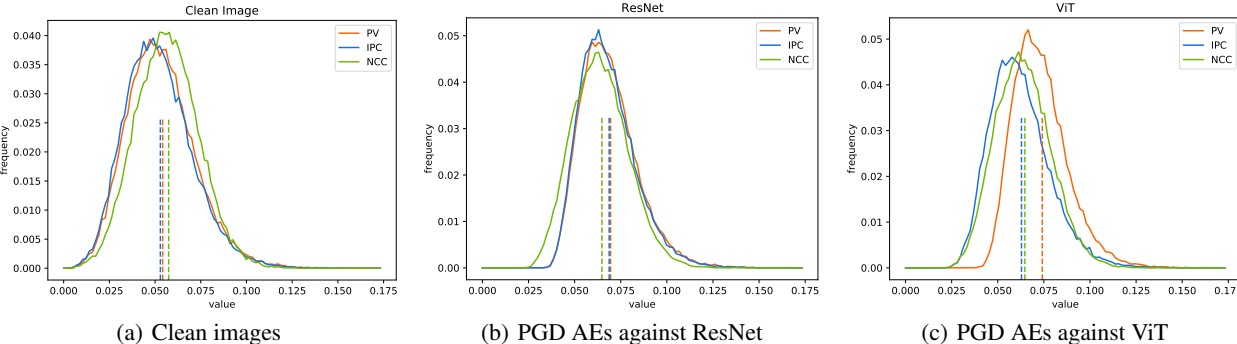

| (a) Clean images | (b) PGD AEs against ResNet | (c) PGD AEs against ViT |

Figure 3: The distributions of the statistics PV, IPC and NCC of the images or AEs on the CIFAR-10 training set. The AEs are generated by the PGD $\ell_\infty = 8$ attack. The victim models are ResNet and ViT respectively. The solid lines in the figures represent the frequencies of the statistics accumulated by 100 groups. The dashed lines are the averages of the according statistics.

clean images. And since PV, IPC and NCC are all statistics, if the the simple linear classifier works well, DNNs should be more capable to dig out the infomation of the AEs in Patch Vestiges.

## Experimental Setups

**Datasets** We use the CIFAR-10 (Krizhevsky 2009) dataset for our experiments. The CIFAR-10 dataset has 50,000 training images, 10,000 test images and 10 categories. The size of each image is 3×32×32.

**Attacks** We use the white-box adversarial attack methods FGSM (Goodfellow, Shlens, and Szegedy 2015), BIM (Kurakin, Goodfellow, and Bengio 2017a), PGD (Kurakin, Goodfellow, and Bengio 2017b) and DeepFool (DF) (Moosavi-Dezfooli, Fawzi, and Frossard 2016). We restrict all the AEs with $\ell_\infty = 8$. Notice that DF originally generates AEs with $\ell_\infty \leq 8$. We rescale the perturbations and use DF* to denote the modification. We run all the BIM and PGD attacks for 20 iterations. We use the AEs of PGD to train our linear regression classifier and directly test it with the AEs from all the attack methods.

**Victim Models** The major victim model is the vanilla Vision Transformer (ViT) (Dosovitskiy et al. 2020). We also use ResNet (He et al. 2016) as a contrast model. ViT used in the experiments has a 4×4 patch size, 6 layers and 16 heads for Multi-Head Attentions. The ResNet model in the experiments has 56 layers.

**Compared Methods** We use LID (Ma et al. 2018) and NSS (Kherchouche et al. 2020) for comparison. Notice that the settings of these methods are not in accord with ours strictly. For example, our classifier only requires the input image, the ViT logits and the patch size. The comparisons are mainly used as a reference.

**Environments** We build our project on the open-source toolbox ARES (Dong et al. 2020) and make references to the codes of TRADES (Zhang et al. 2019). We run the experiments on GeForce RTX 2080 Ti.

## Results

We first compare the distributions and the averages of PV, IPC and NCC between the clean images, the AEs using the PGD attack against ResNet and the AEs against ViT. All the images and AEs are from the training set of CIFAR-10. The results are shown in Figure 3. We observe that for the clean images and the AEs against ResNet, the distributions of PV and IPC are close. But PV of the AEs against ViT is much larger than IPC. We use the T'-test and confirm that PV is significantly larger than IPC and NCC (p<5e-4) in Figure 3(c). We can also observe in Figure 3(c) that there are large area where the distribution of PV is not overlapped with IPC and NCC. The results prove that the assumptions and approximations about $Leaps$ are effective, and illustrate that Patch Vestiges are the unique and significant characteristics of the AEs against ViT.

| Model | KR | DR | | | |
|---|---|---|---|---|---|
| | | FGSM | BIM | PGD | DF* |
| LID | **96.17** | 57.36 | 85.17 | **94.51** | **78.64** |
| NSS | 93.29 | 87.10 | 66.26 | 62.09 | 61.55 |
| Ours | 86.90 | **94.60** | **87.74** | 88.11 | 74.62 |

Table 1: The keep rates (KR, %) of the clean images and the detection rates (DR, %) against the different AEs of the compared models. DF* is the DF modification introduced in the experimental setup section. All the AEs have perturbations with $\ell_\infty = 8$.

We also train a linear classifier (named **Ours** in Table 1) using the PGD attack and compare the keep rates (KR, the ratio of classifying clean images correctly) and the detection rates (DR, the ratio of classifying AEs correctly) under different attacks. Our linear classifier is trained on the CIFAR-10 training set with the PGD attack. All the results in Table 1 are tested on the CIFAR-10 test set. We observe that the simple linear regression classifier, although not state-of-the-art, is comparable enough with the mature adversarial detection methods. This again suggests that Patch Vestiges are significant. The results also show that Patch Vestiges are

the intrinsic attributes of the AEs against ViT and are easily transferred from one attack method to another.

## Q: Is it useful to reduce the perturbated pixels to avoid the effect of Patch Vestiges?

**A:** In some cases, yes.

One major intuitive suspicion about Patch Vestiges may lie in the fact that our method takes all the pixels in the images or AEs into account. To confirm whether reducing the perturbated pixels would reduce Patch Vestiges or not, we make a specific experiment to answer this question. We vary the proportion of the perturbated pixels from 10% to 100%. Two dependent variables are watched: 1. the detection rates of one linear classifier that is trained with 100% pixels perturbated; 2. the detection rates of another linear classifier that is trained with the same percentages of perturbated pixels as the test procedure. The results are shown in Figure 4.

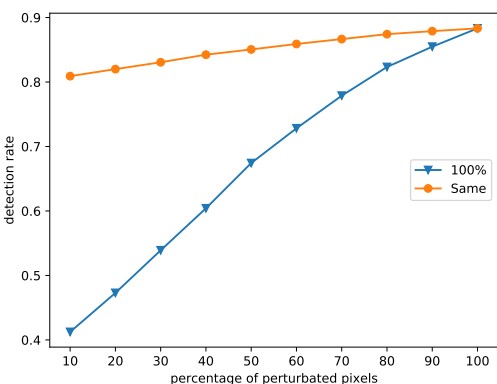

Figure 4: The changes of the detection rates of the two linear classifiers with different percentages of perturbated pixels on the CIFAR-10 test set. "100%": the classifier is trained where all the pixels in the AEs are perturbated. "Same": the perturbated pixels in the training and test procedures have the consistent proportions. Other settings of the classifiers are the same as the above experiments. The AEs are generated by PGD $\ell_\infty = 8$ attack.

We can observe evident declines of the detection rates of the both classifiers when there are fewer perturbated pixels, which makes sense with the intuition. However, the classifier trained with the same proportion declines very slowly. When there only 10% perturbated pixels, the classifier can achieve above 80% detection rate. These phenomena indicate that, when reducing the proportion of the perturbated pixels largely, Patch Vestiges will also be reduced to a great extent, but the remained Patch Vestiges are still large enough for adversarial detection. In view of the fact that reducing the perturbated pixels will possibly reduce the attack performance, this topic may become another focus of attack and defense.

## Conclusion

In this paper, we confirm the human intuition that the division of the patches by Vision Transformer remains large

vestiges in the adversarial examples. We bring up the concept Patch Vestiges to measure to what extend the patches can leave over their traces into the AEs. We also quantitatively show that Patch Vestiges can be leveraged to detect whether an image is an adversarial example against ViT or a clean one.

Besides the practical significance, our work can also promote the thinkings towards the adversarial examples and the AI safety. Is a more complicated structure more vulnerable? The answer is "yes" under many circumstances, considering that most of the new structures are not designed for the security purpose of the models but bring in more potential defects. However, in this paper, we observe that the artifacts of Vision Transformer on the contrary improve its robustness. The attack and defense around the special structures of the models is expected to become a new and attractive topic in the future.

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
