# OpenReview forum: "Patch Vestiges in the Adversarial Examples Against Vision Transformer Can Be Leveraged for Adversarial Detection"
_AAAI.org/2022/Workshop/AdvML — AAAI-22 AdvML Workshop ShortPaper_

### Official Review · Reviewer_oWJ5 · 2021-11-27
**A good research to discriminate adversarial perturbations between ResNet and transformer.**

**Rating:** 6
**Confidence:** 4

**Review:**

Pros:
1. The paper is easy to follow.
2. Utilization of patch vestiges in vision transformer is a new perspective to develop defense models
Cons:
1. The experiments are too simple. More comparisons should be added to comprehensively exhibit the advantages of this method.
2. To demonstrate the generality of the patch vestiges in adversarial perturbations, more adversarial attack methods should be considered.

---

### Decision · Program_Chairs · 2021-12-01

**Decision:**

Accept (Short Paper)

**Comment:**

The reviewer agrees to accept this paper.